# Speculative Decoding with CTC-based Draft Model for LLM Inference Acceleration

**Zhuofan Wen**[1,4]**,Shangtong Gui**[1,2,4]**,Yang Feng**[1,3,4*]
[1]Key Laboratory of Intelligent Information Processing,
Institute of Computing Technology, Chinese Academy of Sciences
[2]State Key Lab of Processors,
Institute of Computing Technology, Chinese Academy of Sciences
[3]Key Laboratory of Al Safety, Chinese Academy of Sciences
[4]University of Chinese Academy of Sciences, Beijing, China
{wenzhuofan24z,guishangtong21s,fengyang}@ict.ac.cn

## Abstract

Inference acceleration of large language models (LLMs) has been put forward in many application scenarios and speculative decoding has shown its advantage in addressing inference acceleration. Speculative decoding usually introduces a draft model to assist the base LLM where the draft model produces drafts and the base LLM verifies the draft for acceptance or rejection. In this framework, the final inference speed is decided by the decoding speed of the draft model and the acceptance rate of the draft provided by the draft model. Currently the widely used draft models usually generate draft tokens for the next several positions in a non-autoregressive way without considering the correlations between draft tokens. Therefore, it has a high decoding speed but an unsatisfactory acceptance rate. In this paper, we focus on how to improve the performance of the draft model and aim to accelerate inference via a high acceptance rate. To this end, we propose a CTC-based draft model which strengthens the correlations between draft tokens during the draft phase, thereby generating higher-quality draft candidate sequences. Experiment results show that compared to strong baselines, the proposed method can achieve a higher acceptance rate and hence a faster inference speed.

## 1 Introduction

Large Language Models (LLMs) have been applied to a wide range of text generation tasks such as machine translation and question answering due to their remarkable performance[1, 21, 6, 11]. In many applications, LLMs is required to produce a long generation to give explanations to its answer or perform chain of thought (CoT). Moreover, in some practical scenarios, LLMs have to resort to other applications to fulfill a task under the frame of AI agents where LLMs usually generate long outputs to communicate with other applications. All of these have a high demand for the inference speed of LLMs. However, most LLMs employ a token-by-token autoregressive generation paradigm, bringing on the severe inference delay problem, which obstacles the real applications of LLMs.

To mitigate the inference latency of LLMs, speculative decoding is proposed and proves to be a more efficient decoding strategy compared with autoregressive generation[12, 4]. Besides the base LLM, speculative decoding usually introduces a drafter model in the working flow that the draft model generates candidates for the next several tokens and the base LLM verifies the candidate and decides to accept or reject at some criterion. Once accept, then the winner candidate will be used as the output, otherwise, the base LLM will decode and generate the output. In this process, the draft

---

[*]Corresponding author: Yang Feng

38th Conference on Neural Information Processing Systems (NeurIPS 2024).

model usually has few parameters and hence produces generations at a faster speed. Meanwhile, the LLM can verify draft tokens parallelly and takes less time than generating the next several tokens by itself. As only the candidate is accepted at a high rate, the decoding speed can be improved. It can be derived that the final inference speed is related to the decoding speed of the draft model and the acceptance rate of the candidate selected out by the draft model. Nevertheless, there is a trade-off for the draft model between its decoding speed and its performance related to acceptance rate.

Following the principle of speculative decoding, many works focus on optimizing the draft model to accelerate inference [15, 19, 5, 17] in which the methods based on non-autoregressive (NAR) generation have shown promising results. The NAR speculative decoding methods draft the next several tokens parallelly by predicting them independently based on the representation of the original LLM. Although these methods drafts at a high speed, they ignore the dependence between the next several tokens and sacrifice the performance as the cost. As a consequence, the speed of speculative decoding can be affected via the acceptance rate.

Based on these observations, we make efforts from the perspective of model performance by introducing dependency relationships during draft generation, aiming at achieving a higher acceptance rate. At this end, we propose a draft model based on Connectionist Temporal Classification(CTC) algorithm [9] which generates drafts in a non-autoregressive way with additional blank and repetitive tokens participating in. As during training the CTC-based draft model will count all the possible candidates sequentially that can generate the given ground truth when calculating the probability of the ground truth, the candidates with better dependency relationships will achieve higher probabilities. As a result, at inference the best candidate selected out by the CTC-based draft model will be more sequentially reasonable and hence can be accepted at a higher rate, ultimately leading to faster inference. For the rest of this paper, we refer to CTC-drafter as CTC-based draft model for short. Experiments on MT-bench show that the proposed method is able to draft sequences at a higher acceptance rate compared to strong baselines, thus achieving remarkable inference speedup.

Our contributions are as follows:

1. We introduce the CTC-based draft model to speculative decoding framework, to the best of our knowledge, which is the first to apply the CTC algorithm within the speculative decoding domain. This approach can not only generate drafts in a non-autoregressive way but also introduce correlations between draft tokens through probability allocation.

2. Through experiments conducted on MT-bench and GSM8K using various LLMs as base models, we have demonstrated superior speedup ability of the CTC-based draft model compared to other speculative decoding improvement methods. These results prove the rationality and effectiveness of our method.

## 2   Background

Recent advancements have emerged from the innovative approach of Blockwise Decoding[17], which introduced the draft-then-verify paradigm, leading to the development of Speculative Decoding[12] and Speculative Sampling[4]. These methodologies offer promising avenues for enhancing the speed of Large Language Models (LLMs).

Speculative Decoding predicts multiple future tokens and verifies their accuracy within a single decoding step. Using greedy sampling as an illustration: at step $t$, given an initial prompt $X$ and previously produced tokens $y_{<t} = y_1, ..., y_{t-1}$, a speculative sequence of length $n$, $y'_t, ..., y'_{t+n}$, is generated by the draft model with respective probabilities $p'_t, ..., p'_{t+n}$. The target LLM then computes the accurate probabilities $p_t, ..., p_{t+n}$ in one pass during verification. Each token $y'_i$ is evaluated in sequence, with its acceptance probability given by $\min(1, p'_t/p_t)$. Upon rejection of a token $y'_i$, subsequent tokens are disregarded, and the rejected token is re-sampled using the adjusted distribution $P(y_i) = \mathrm{norm}(\max(0, P(y_i|y_{<i}, X) - P'(y_i|y_{<i}, X)))$.

The effectiveness of Speculative Decoding significantly depends on designing an intelligent draft model for precise token prediction and devising an optimal strategy for token sequence verification[24]. Consequently, current research efforts concentrate on these aspects to further exploit the potential of Speculative Decoding for speed acceleration.

## 2.1 Design of Draft Model

Many researchers have pursued the strategy of designing a draft model that operates independently of the base model[15, 19, 5, 13], such as employing a non-autoregressive transformer for simultaneous token drafting[23]. To ensure compatibility with the base model, works like [12] opt for draft models with fewer parameters from the same model series.

However, independent draft models necessitate training or fine-tuning, posing flexibility issues when transitioning between base models. Alternatively, some approaches rely on modifying the base model itself for token drafting through moderate adjustments[3, 25, 26, 10]. For instance, [3] introduces an additional module comprised of linear layers atop the target LLM for drafting tokens independently for different positions. In contrast, [25] incorporates a bypass within the LLM, allowing for earlier exits during the model's layer-by-layer computation.

## 2.2 Optimization of Verification

The strategy for compiling drafted tokens into candidate sequences and the criteria for sequence selection are vital during the verification stage. Initially, forming a single candidate sequence from the most probable tokens across positions was the prevalent approach[16, 18]. To incorporate a broader range of draft sequences, SpecInfer[14] organizes draft tokens into a tree structure, with paths from the root to leaf nodes representing different candidate sequences. Regarding selection criteria, early methods only accepted sequences matching the target model's greedy decoding output[23]. Later, [4] introduced Nucleus Sampling as a more effective yet complex acceptance criterion.

## 2.3 Connectionist Temporal Classification

Connectionist Temporal Classification (CTC) is tailored for sequence prediction tasks, especially applicable in speech and handwriting recognition[9]. CTC expands the output space, $\mathcal{Y}$, by introducing a blank token $\epsilon$ denoting 'output nothing', creating an augmented space $\mathcal{Y}^*$. It defines a function $\beta(y)$ that maps any sample $y \in \mathcal{Y}$ to a subset of $\mathcal{Y}^*$, containing all valid alignments. Conversely, $\beta^{-1}$ processes alignments from $\mathcal{Y}^*$ back to $\mathcal{Y}$ by merging adjacent, repeated tokens and removing blanks, resulting in the target sentence.

The sequence-level CTC loss function offers superior context modeling capabilities compared to token-level alternatives and effectively manages variable-length outputs without necessitating alignment during training. The training objective leverages dynamic programming to aggregate over all potential alignments $a \in \mathcal{Y}^*$.

$$\log p(y) = \log \sum_{a \in \beta y} p(a) \tag{1}$$

For inference, the model generates alignment $a$, from which repeated tokens and blanks are removed to yield the final output $y = \beta^{-1}(a)$.

# 3 CTC-drafter Model

In this section, we describe our implementation of the proposed model. CTC-drafter improves the acceptance rate of draft tokens while keeps extra investment of draft time in a reasonable duration, consequently achieving superior inference speedup. We first give an illustration of CTC-drafter's model structure, analyzing the functions of involved modules. Subsequently, we clarify CTC-drafter's training strategies and inference procedure.

## 3.1 Model Structure

Our CTC-drafter model structure are displayed in Figure 1. The training strategy is on the upper part and the inference process is on the bottom part. The base model on the left side are the LLM we desired to accelerate, which generally is composed of an embedding layer, multiple attention transformer layers and a output LM head that maps the hidden states to probability in vocabulary dimension.

For the draft procedure, we insert an attention draft module which take the hidden states outputted from base model as input and predict the probability distributions of draft tokens. Here hidden states

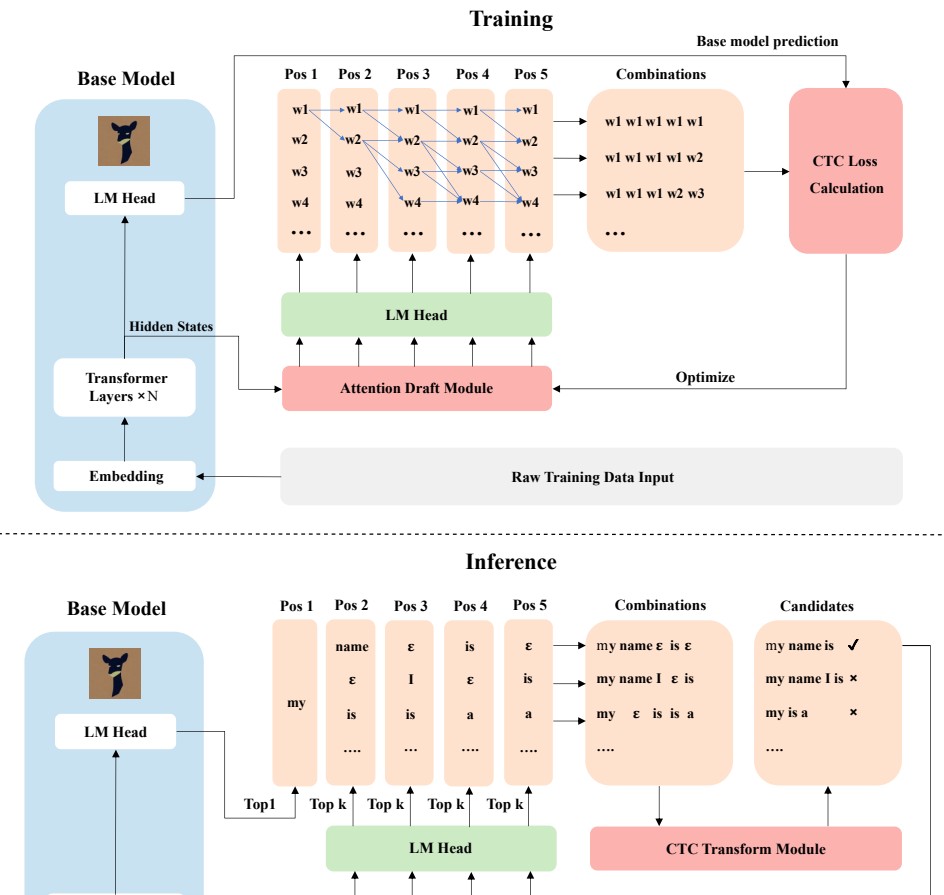

Figure 1: Illustration of CTC-drafter model training and inference strategy.

represents the compressed feature sequence after multiple transformer layers. Inspired by [13], the inner construction of Attention Draft Module is similar to the base model, with one single transformer layer conducting prediction in parallel.

For the verify procedure, raw candidate sequences are acquired by combining draft tokens in different places. Different from Medusa[3] which cuts off a part of combinations as prearranged, all the raw sequences keep the same length, containing possible draft tokens in each place. A CTC Transform Module is designed to process all raw sequences. The module first removes consecutive duplicate tokens and blank character, namely $\varepsilon$ in Figure 1. Then the attention map that is used in base model verification calculation is modified. Positions in the attention map that corresponds to tokens been removed in CTC transform will be masked.

## 3.2 Training

The basic training strategy of CTC-drafter is displayed in Figure 1. We fixed the parameters of base model and trained the transformer layer in Attention Draft Module on ShareGPT dataset, which

is a subset of Vicuna's[6] training data. All input sequences are padded to the same max length, which is a predefined hyper parameter. Instead of conventional token-level cross entropy function that separately calculate the loss of each position, we use sequence-level CTC loss as the training objective. Note the input sequence as $X$ and its corresponding label as $Y$, The dataset $D$ contains a set of $(X, Y)$ pair. The parameters of the trained draft model is noted as $\theta$. Our training objective is optimizing $\theta$ to maximize the probability of labels across the dataset:

$$\theta = \underset{\theta}{\mathrm{argmax}}(\sum_{(X,Y)\in D} P(Y|X,\theta)) \tag{2}$$

Conventionally, the training labels can be acquired by simply shifting the input. To train CTC-drafter, we follow the knowledge distillation method to calculate $Y_{distill}$ as labels $Y$ in equation 2 by inputting base model the origin data [28], for the consideration that draft model can better match base model if trained on distilled dataset. First, base model outputs the probability distribution $P_{distill}(Y|X)$ in equation 3 and equation 4 trough multiple transformer layers and LM head, then we get distilled label sentence $Y_{distill}$ by greedy decoding in equation 5:

$$F_{distill} = \mathrm{BaseModel}(X) \tag{3}$$

$$P_{distill}(Y|X) = \mathrm{Softmax}(\mathrm{LmHead}(F_{distill})) \tag{4}$$

$$\mathrm{Y}_{distill} = \underset{Y}{\mathrm{argmax}}(P_{distill}(Y|X)) \tag{5}$$

Given labels, we use sequence-level CTC loss to model equation 2. Assume the set $A_{X,Y}$ contains all possible raw sequences $A$ that can be converted to $Y$ after removing reduplicate tokens and blank characters. Sum up all probabilities of sequence in the set to get $P(Y|X,\theta)$. Considering the time complexity, it is impractical to enumerate all sequence. In Figure 1, we briefly show the dynamic programming of traversing all routes to calculate training labels probability $P(Y|X,\theta)$ in equation 6. The detailed algorithm is discussed in [9].

$$P(Y|X,\theta) = \sum_{A\in A_{(X,Y)}} P(A|X,\theta) \tag{6}$$

Further, the probability of each sequence $P(A|X,\theta)$ equals to the product of probability of each token $a_i$ in the sequence with the independent assumption:

$$P(A|X,\theta) = \prod_{t=1}^{T} p(a_t|X,\theta) \tag{7}$$

$$A = (a_1, a_2, a_3, \ldots, a_t, a_{t+1}, \ldots, a_n) \tag{8}$$

The probability distribution of each token in the sentence is predicted by the transformer layer we added in draft module based on the hidden states outputted from base model as in equation 3:

$$\hat{F} = \mathrm{DraftModule}(F_{distill}) \tag{9}$$

$$\hat{P}(Y|X,\theta) = \mathrm{Softmax}(\mathrm{LmHead}(\hat{F})) \tag{10}$$

$\hat{P}(Y|X,\theta)$ actually contains a group of probability distribution of different places in the sentence. Locate the corresponding position $\hat{P}_t(Y|X,\theta)$ to calculate $p(a_t|X,\theta)$ in equation 7:

$$p(a_t|X,\theta) = \hat{P}_t(Y = a_t|X,\theta) \tag{11}$$

### 3.3 Inference

To clarify the inference speedup mechanism of CTC-drafter, we further explain this procedure in one specific decoding step with the decoding history "Usr: what is your name? Assistant: hello," as base model input. The input will first be passed through base model producing the hidden states and greedy sampling base token "my". Attention Draft Module takes the hidden states from last transformer layer as input, outputting probability distributions of different positions after base token after LM Head projection.

For every position, the top k tokens are selected in descending order of probability, where k is predefined. In this instance, Attention Draft Module suggests that "name" is the best candidate token

Table 1: performance of average speedup ratio on MT-bench.$\gamma$ represents the average speedup ratio for all evaluation questions relative to Vanilla method, calculated by equation 13. $\beta$ represents the average number of accepted tokens per decoding step for all evaluation questions, calculated by equation 12.

| Speculation Method | Vicuna-7B | | Vicuna-13B | | Vicuna-33B | |
|---|---|---|---|---|---|---|
| | $\gamma$ | $\beta$ | $\gamma$ | $\beta$ | $\gamma$ | $\beta$ |
| MT-bench | | | | | | |
| Vanilla[6] | 1.00× | 1.00 | 1.00× | 1.00 | 1.00× | 1.00 |
| Medusa[3] | 2.13× | 2.58 | 1.97× | 2.60 | 1.93× | 2.55 |
| Hydra[2] | 2.36× | 3.04 | 2.17× | 3.06 | 2.15× | 2.95 |
| CTC-drafter | **2.78×** | **3.56** | **2.52×** | **3.51** | **2.20×** | **3.53** |
| GSM8K | | | | | | |
| Vanilla[6] | 1.00× | 1.00 | 1.00× | 1.00 | 1.00× | 1.00 |
| Medusa[3] | 2.33× | 2.78 | 2.21× | 2.68 | 2.10× | 2.46 |
| CTC-drafter | **2.43×** | **3.53** | **2.66×** | **3.53** | **2.16×** | **3.40** |

right next to "My" while it is possible that a blank character appears after "my" and the first draft token. Attempting to cover more reasonable candidate sequences, tokens in each place with different probability are combined in token tree structure and a group of the most valuable combinations are reserved as the raw candidate sequences. The raw candidate sequences is refined in CTC Transform Module, removing repetitive tokens and blank character and modifying the attention map.

All candidates are verified parallelly in base model, the longest sequence that satisfies the criterion will be selected as current decoding step's output, which in this case is "my name is". The decoding history is updated according to the output. In one single decoding step, compared with autoregressive decoding which will only produces token "My", CTC-drafter enables multiple tokens to be outputted, thus reducing overall base model calculation steps and achieving speedup.

## 4 Experiments

### 4.1 Implementation Settings

We choose open-source Vicuna large language model[6] with different parameter sizes as base model to conduct experiments. Vicuna models is fine-tuned on ShareGPT dataset based on LLaMA model, which are noted below as Vicuna-7b, Vicuna-13b and Vicuna-33b according to different parameter sizes. We also conduct training on LLaMA-2-Chat base models, detailed in the Appendix.

We fix Vicuna model's parameters and train the transformer layer inside draft module on ShareGPT dataset. The learning rate is set to $3 \times 10^{-5}$. To avoid gradient explosion, we adopt gradient clipping, setting the clipping threshold to $0.5$. We set the max length of training data to $2048$. All training tasks were executed on four 24GB NVIDIA GeForce RTX 3090 devices, taking around two days. To fully utilize graphics memory and accelerate training, we load models with FP16 precision for quantization. For comparison, we also implemented Medusa[3] on Vicuna models, following suggested experiment settings and retrain on the same ShareGPT dataset.

Trained models are evaluated on MT-bench and GSM8K datasets to assess the acceleration performance in various scenarios. MT-Bench is a carefully curated benchmark that includes 80 high-quality, multi-turn questions covering 8 primary categories of user prompts such as writing, roleplay and extraction[27]. GSM8K contains 8.5K high quality linguistically diverse grade school math problems[7]. Unlike some other datasets that offer base model questions with definitive answers such as multiple-choice questions, the two selected evaluation datasets contain open-ended questions, requiring base model to output long sequence answers in multiple decoding steps.

For every question, we record the total number of tokens in its corresponding answer as $N$, the total inference time $T$ and the base model decoding steps $M$. We calculate the average number of tokens accepted per decoding step and the inference speedup compared to vanilla base model without

Table 2: Performance of average speedup ratio on MT-bench for different model structures. $\gamma$ represents the average speedup ratio for all evaluation questions relative to Vanilla method, calculated by equation 13. $\beta$ represents the average number of accepted tokens per decoding step for all evaluation questions, calculated by equation 12.

| | CTC Verify | | Medusa verify | |
| --- | --- | --- | --- | --- |
| Speculation method | $\gamma$ | $\beta$ | $\gamma$ | $\beta$ |
| Linear layer + Cross Entropy Loss | 1.71× | 2.38 | 2.13× | 2.58 |
| Transformer layer + CTC Loss | 2.78× | 3.56 | 2.25× | 3.02 |

speculative decoding:

$$\text{Accepted tokens} = \frac{N}{M} \tag{12}$$

$$\text{speedup} = \frac{\bar{T}_{\text{vanilla}}}{\bar{T}_{\text{spec}}} = \frac{T_{\text{vanilla}}/N_{\text{vanilla}}}{T_{\text{spec}}/N_{\text{spec}}} \tag{13}$$

The average number of accepted tokens reflects models' ability to speculate candidate tokens. However, it takes extra inference time to draft tokens. Thus, the speedup metrics can be viewed as a trade-off between speculation quality and extra time consumed.

## 4.2  Results and analysis

The performance of different speculation methods on MT-bench and GSM8K are showed in Table 1. The speedup ratio of Medusa is evaluated following recommended setting in its corresponding technical report. The results of Hydra[2] on Vicuna models are acquired from its corresponding paper. We also measures the performance of fully auto-regressive decoding with no speculation method as baseline to calculate speedup ratio of other three methods, noted as Vanilla in Table 1.

**MT-bench.** The results show that our proposed CTC-drafter achieves better draft quality on MT-bench compared other works, with more than three tokens been accepted per decoding steps. Higher predicting accuracy enables CTC-drafter to achieve speedup ratio of more than 2×, outperforming Medusa and Hydra on all types of base model. Besides, the speedup performance of all speculation method is influenced as base model size increases. Possible explanation is that we can not significantly expand the size of draft module considering extra time consumed, when base model size increase, larger ability gap between base model and draft module makes it more difficult for draft module to imitate base model's prediction behavior. Therefore, the average number of accept tokens $\beta$ of CTC-drafter decreases from 3.56 to 3.53, influencing the speedup performance.

**GSM8K.** Compared with MT-bench, questions in GSM8K mainly focus on math category. As is shown on the bottom of Table 1, our proposed CTC-drafter keep a superior speedup performance over Medusa for all base models. Besides, the speedup performance suffers to some extent for CTC-drafter in Vicuna-7B base model, compared with the performance in MT-bench. The main reason is that we completely rely on the comprehensive ability of origin base model to offer answers without fine-tuning on GSM8K training dataset. Fortunately, when the base model size increases to 13B, CTC-drafter maintains prediction accuracy, achieving 2.66× speedup. However, bridging the capability gap for Vicuna-33B is challenging, leading to a decline in performance.

## 4.3  Ablation experiments

In this part, we list the results of ablation experiments and further analyze the working paradigm of CTC-drafter. First, we explore how each part of the model structure influences the acceleration performance in Table 2. Then we illustrate how the prediction ability varies across different categories of test questions in Figure 2. Besides, to better visualize the trade-off between extra time consumption and prediction accuracy, the time consumption of each calculation procedure during inference is measured in Figure 3.

**Model structure**. To better utilize the context information, CTC loss is used as the training objective as discussed in Section 3.2, which optimizes the draft module under sequence-level supervision. Besides, we replace the linear layers of Medusa head with more complex transformer layer to suit

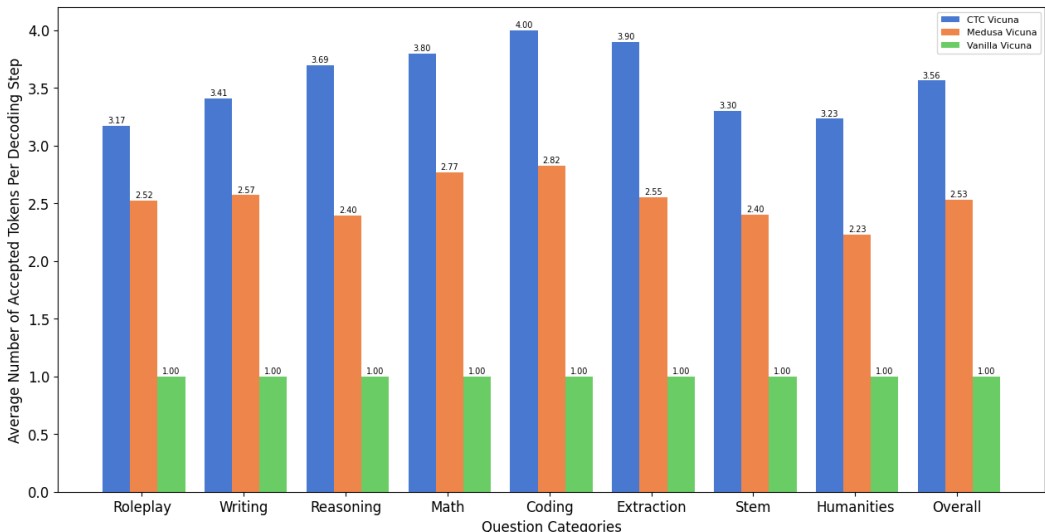

Figure 2: Average number of tokens accepted per decoding step in different question categories on MT-bench, with Vicuna-7B as base model. The performance on Vicuna-13B and Vicuna-33B is consistent with this result. The blue color represents CTC-drafter method, orange color represents Medusa method and green color represents baseline. All evaluation experiments are conducted on the same device.

CTC loss and better imitate the base model. For inference, the original token tree verification strategy is modified to include extra operations such as CTC transform and attention map modification.

To explore the speedup contribution of each part of modification as discussed above, we replace the draft module and verify module with the corresponding ones in Medusa and conduct experiments on the modified speculation methods. The results are showed in Table 2, with Vicuna-7B as base model. In this table, linear layer represents drafting tokens based on linear layers with cross-entropy Loss function as training objective, which is adopted by Medusa. Transformer layer represents drafting tokens based on transformer layers with CTC loss as training objective, which is adopted by CTC-drafter. Medusa verify refers to vanilla token tree verification described in [14]. CTC verify includes extra operations compared with Medusa verify including CTC transform of candidate sequences and attention map modification as mentioned before.

Replacing linear layer with transformer layer and using CTC loss to design training loss function increase the average number of accepted tokens $\beta$ from 2.58 to 3.02. It is clear that with these two efforts combined, draft module is guided to conduct attention across the whole input sentence instead of simply learning offsets of the last hidden states. However, blank characters and repeated tokens exist in the candidate sequences spoil draft quality and speedup for models without CTC transform, $\beta$ decreases from 3.56 to 3.02 and $\gamma$ from 2.78× to 2.25×.

**Question categories**. We further explore how speedup performance varies on different question categories, showing in Figure 2. Both CTC-drafter and Medusa achieved the best prediction accuracy on coding category, which can be attributed to the highly logical nature of the problems within this category. Among all categories, the acceptance rate of roleplay questions is slightly low for CTC-drafter, which may be due to the deficiency of questions of this category in our training datasets.

**Time consumption**. Compared with Medusa, it is unavoidable that our methods' draft strategy requires more complex calculations. We display each stage's time consumption throughout the whole inference decoding process in Figure 3. First, we replace the original medusa head with transformer layer to better fit the base model, which cause the time of draft model increases from 3.71% to 14.93%. Besides, for the need to dynamically process candidate sequences in each decoding round, the CTC transform accounts for extra 5.36% of the overall decoding time consumption. Considering that the base model's calculation still account for the main part, it is acceptable that we increase the draft ability and thus reducing base model's decoding rounds, which balances the extra time consumption and achieve better speedup on the whole.

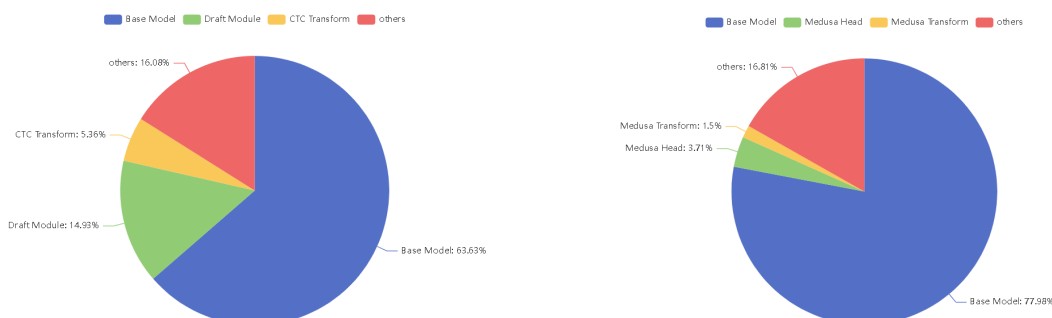

Figure 3: The percentage of time consumed for different processes based on CTC-drafter(left) and Medusa(right) speculation strategies. The "others" part mainly contains matrix operations involved in token tree verification.

## 5 Related Work

**Medusa.** After Speculative decoding[23] and Speculative sampling[12], many improvement works has been proposed to optimize the draft model and verification strategy. Among these, Medusa explores a novel and efficient acceleration framework[3]. Instead of using an independent model with fewer parameters as the draft model, several Medusa Heads are added on top of the last transformer layer of base model. The i-th Medusa Head is responsible for predicting the i-th token after the base model decoding token in each step. For each Medusa Head, top k tokens with highest probability are selected and combined in tree structure to form candidate sequences. Using tree mask method in [14], the multiple sequences are validated in parallel during the next decoding step. Two strategies are used for the training of Medusa Head: Medusa-1 fix the parameters of base model, optimize only the Medusa Head on a small dataset, Medusa-2 adopt end-to-end training, fine-tuning base model and Medusa Head together on larger datasets. Although Medusa-2 can achieve remarkable inference speedup, the need of large training data and time consumption limits its generality across different base model. In this paper, we only implement and demonstrate Medusa-1 considering fairness.

**Hydra.** Modified from Medusa Head, Hydra designs Hydra Head, a sequentially dependent, drop-in replacement for standard draft heads[2]. A Hydra Head conduct prediction not only based on base model hidden states but also the decoding output of other Hydra Heads, which significantly improves speculation accuracy. Besides, some other tricks are adopted for further inference speedup including adding noise and knowledge distillation.

## 6 Conclusion and Future work

Speculation decoding and corresponding improvement works mostly draft candidate tokens without considering context information and generate fixed-length candidate sequences for verification, which not only influences the draft quality, bur also lacks generality across different large language models. In this paper, we propose a novel framework named CTC-drafter based on CTC algorithm. Specifically, we use CTC loss as the training objective to model the context connection instead of cross entropy. We reconstruct the structure of draft model, using transformer layer to better fit base models. Besides, with CTC transform, we achieve adaptive candidate sequence generation which makes it convenient to transfer the framework across different base models. Nevertheless, our current work is subject to certain limitations that requires careful consideration. More training tricks can be explored to further enhance the prediction ability of draft module. Besides, it is still doubtful that whether the current draft model structure is optimal. What's more, different types of large pretrained language model need to be adopted in our proposed framework to evaluate the acceleration performance of CTC-based draft model.

For the future work, we attempt to identify techniques to reduce the extra time consumed caused by more complex draft operations introduced. Other verification criteria such as Nuclear Sampling [4] can be integrated into our framework. What's more, some other methods such as Conditional Random Field(CRF, [20]) and Directed Acyclic Graph(DAG, [8]) can be explored to model the context information when drafting tokens, we remain these ideas for future work.

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

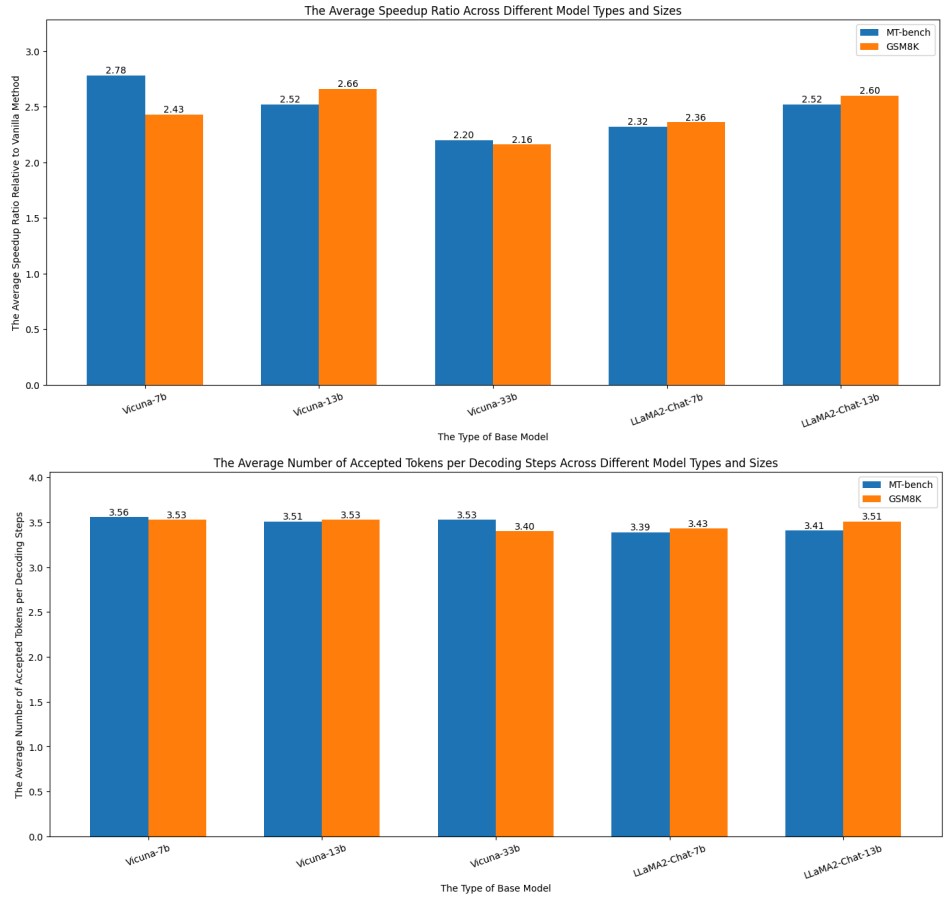

Figure 4: The bar charts of speedup ratio relative to vanilla method $\gamma$(top) and average number of tokens accepted per decoding step $\beta$(bottom) across different model types and sizes with CTC-drafter. The blue bar represents performance on MT-bench and the orange bar represents GSM8K. All evaluation experiments are conducted on the same devices.

## A   Appendix

To evaluate the generality of our method across different base models, we add supplementary experiments on LLaMA-2-Chat base models[22]. We select LLaMA-2-Chat 7b and 13b as base models and evaluate CTC-drafter's performance on MT-bench and GSM8K. For a clear comparison, the evaluation results of various base models, including Vicuna, are documented in Figure 4.

CTC-drafter maintains ideal performance when transferring from Vicuna models to LLaMA-2-Chat models, only slight decline when compared the evaluation results on Vicuna-7b and LLaMA-2-chat-7b. Besides, it should be noted that increasing the size of the LLaMA-2-Chat model to 13b does not compromise draft quality, while enhancing speedup performance. This trend diverges from Vicuna base models, potentially due to distinct inference paradigms inherent in both models.

