# OpenReview forum: "Speculative Decoding with CTC-based Draft Model for LLM Inference Acceleration"
_NeurIPS.cc/2024/Conference — NeurIPS 2024 poster_

### Official Review · Reviewer_NRvU · 2024-07-11

**Soundness:** 3
**Presentation:** 3
**Contribution:** 3
**Rating:** 6
**Confidence:** 3

**Summary:**

The paper proposes a novel architecture and training technique for LLM speculative decoding, aiming to improve the reliability and acceptance rate of generation candidates. Unlike Medusa, the proposed method replaces the draft module with a Transformer and utilizes CTC-based loss instead of CE loss. For CTC training, pseudo-labels generated from the base model, rather than ground-truth tokens, are used. During inference, the draft head generates probability candidates, and the CTC beam search process produces the final candidates for evaluation. Experimental results show that the CTC-drafter achieves higher speedup due to an increased acceptance rate and a greater number of accepted tokens.

**Strengths:**

* Combining CTC loss with sequence prediction in text-only LLMs is an interesting approach that could inspire further research.
* The results suggest that CTC-drafter is promising in terms of inference acceleration without significant overhead.

**Weaknesses:**

* CTC is known to suffer from the conditional independence problem, i.e., each token's prediction does not depend on other tokens. Given this, it is unclear how CTC can address the low acceptance rate. The Hydra paper emphasized the importance of token-level sequential dependency, but CTC loss does not seem to support this due to its conditional independence. This is a critical point for the paper’s motivation, so a thorough justification of CTC loss is necessary.
* Is the CTC output decoding during inference the same as the well-known CTC prefix beam search? If so, it should be clarified that this is known as prefix search decoding.
* There is no ablation study on (Transformer Layer + CE loss) or (Linear layer + CTC loss). It is unclear which changes contribute more to performance improvements.

**Questions:**

* How many beams are used for CTC decoding? Is the number of beams (or candidates) the same as in Medusa, ensuring a fair comparison?
* The architecture of the Attention Draft Model is somewhat difficult to understand. Does it take a single hidden embedding vector as input? Is this vector expanded (duplicated? or repeated?) to create an input sequence for the Transformer-based draft model? What is the input of the Transformer-based draft model, and are positional encodings inserted at the beginning?
* The explanation of the CTC-related part could be improved. For example, it would be helpful to mention that different candidates can be expanded to multiple "alignments" before the CTC blank-collapse.

**Limitations:**

The paper adequately addresses the limitations.

---

> ### Author Rebuttal · Authors · 2024-08-06
>
> Many thanks for the insightful comments and constructive suggestions.
>
> $\textbf{R1.Ablation study of draft model structure. (W3)}$
>
> Thank you for the constructive suggestions. We add ablation experiments with modified draft model structure (Transformer Layer + CE loss) and (Linear layer + CTC loss) to evaluate which part contribute more to performance improvements.
> | Draft model structure | Speedup $\beta$ | Token acceptance rate $\gamma$ |
> |:------------------|:--------------------:|:-------------------:|
> | Linear layer + Cross Entropy Loss + CTC Verify      | 1.71×                          | 2.38                                  |
> | Linear layer + Cross Entropy Loss + Medusa Verify     | 1.94×         | 2.53                         |
> | Transformer layer + CTC Loss + CTC Verify     | 2.99×         | 3.73                         |
> | Transformer layer + CTC Loss + Medusa Verify    | 2.25×         | 3.02                         |
> | Linear layer + CTC Loss + CTC Verify   | 1.52×         | 2.07                         |
> | Transformer layer + Cross Entropy Loss + Medusa Verify        | 1.98×         |  2.77                        |
>
> Two observations are listed as follow:
> 1. Based on the results of evaluation on (Linear layer + Cross Entropy Loss + Medusa Verify) and (Linear layer + CTC Loss + Medusa Verify): CTC loss could not match well with linear layer and obvious performance decrease happens.
> 2. Based on the results of evaluation on (Linear layer + Cross Entropy Loss + Medusa Verify) and (Transformer layer + Cross Entropy Loss + Medusa Verify): replacing linear layer with transformer layer could contribute to more precise draft prediction while more complex calculation of transformer layer obstructing the overall speedup.
>
> $\textbf{R2. Discussions about the conditional independence problem of CTC (W1 and Q3).}$
>
> Although the draft model with CTC loss conducts prediction independently in a non-autoregressive manner at each time step, that is also known as the conditional independence, during training it employs dynamic programming to count all the possible generated candidate sequences which can derive ground truth and then adjust the probability distribution to make these possible candidates distributed the greatest probability. In this way, the generated candidate that can derive ground truth most likely, which can also be thought to have the most reasonable sequence dependency, will be selected out by the draft model and meanwhile will be accepted by the base model at a higher rate.
>
>
> The motivation that we combine CTC algorithm into speculation prediction for text-only LLMs is based on the observation that Medusa actually adopt a fully non-autoregressive decoding strategy to generate candidates: Each Medusa head independently generates tokens of the same position without depending on tokens from other medusa heads. During training, it only uses cross-entropy loss as the training objective which only performs word-level matching and cannot introduce sequence-level dependency, hence the quality of the draft is affected. In contrast, CTC-loss introduces sequence-level dependency via adjusting probability distribution, so it provides a better solution to the conditional independence problem existing in Medusa.
>
>
> $\textbf{R3. Detailed explanation of decoding method (W2 and Q1).}$
>
> For the the decoding of CTC-drafter, we first generate original sequences based on the Token-tree used in Medusa, then conduct CTC blank-collapse on the generated original sequences to acquire candidate sequences of different lengths. Here CTC-drafter and Medusa keep the same number of 42 candidate beams, ensuring a fair comparison. This decoding method can better fit the scenarios of speculation decoding.
>
>
> We've tried prefix beam search to generate candidate sequences at first. However, this decoding method is time-consuming, reducing the inference speedup dramatically. What's more, the sequences generated from prefix beam search lack diversity, which leads to relatively low acceptance rate.
>
> $\textbf{R4.Explanation of architecture of the Attention Draft Model (Q2).}$
>
> The internal structure of the Attention Draft Module is similar with that of the base model, using a single transformer layer that takes the hidden states from the last transformer layer of the base model as input and performs prediction in parallel.
>
> Currently, we do not consider expanding the vector. Original sequences are generated based on a token-tree, which is the same as Medusa, and then candidate sequences of varying lengths are obtained after CTC blank-collapse.
>
> The rotary positional encodings are inserted at the beginning of the draft module, following the base model’s configuration.

---

> > ### Comment · Reviewer_NRvU · 2024-08-09
> > **Thank you for the response**
> >
> > Thank you for the response and additional experiments.
> >
> > I hope the discussion about CTC independence and beam search method to be included in the revised version.
> >
> > Regarding the Attention Draft Model's architecture, my confusion is this: Medusa uses different heads for each position (i+1, i+2, ...). When you are saying that the Transformer is used for Attention Draft, is it mean that each Medusa head is a Transformer or all heads are integrated into a single Transformer?
> >
> > * If Former (per-position): Query is only a single timestep input. Do you need a Transformer for this non-sequential input?
> > * If Latter (unified): Is Attention Draft Module Auto-regressive? How can it generate all tokens at once?

---

> > > ### Author Response · Authors · 2024-08-11
> > > **Thank you for the further discussion**
> > >
> > > Many thanks for the further valuable suggestions. We will definitely include the discussion of CTC independence and beam search in the revised version. We would like to give reorganized description about our method to help understand the Attention Draft module’s architecture.
> > >
> > > Overall, the architecture of our method can be decomposed into three parts: the base model, the drafter which includes the Attention Draft module and LM head, and the CTC-related module which is used to calculate CTC-loss for training and performs CTC-style decoding at inference. The main structures of the three parts are as follows.
> > > - The base model in our method is based on the autoregressive generation framework like Llama which only uses Transformer decoder as the main structure.
> > > - For the drafter, the Attention Draft module employs one Transformer layer as its structure including masked multi-head self-attention sublayer and Feed Forward sublayer. Compared to Medusa which employs FFN as the structure of each Medusa head without interacting between Medusa heads, our method can introduce sequence correlations between the preceding generated words and next several words to generate, besides the CTC-related module.
> > > - The CTC-related module involves non-autoregressive generation of the next N tokens (N  is a hyperparameter, which is set to 4 for current CTC-drafter) where blank character and repetitive tokens are introduced into the raw generated sequence which will be processed by CTC Transform module to produce the final draft for the current decoding timestep.
> > >
> > > Our method works in a different way during training and inference. During training, the base model will generate the whole sequence in an autoregressive manner which is used as the ground truth to distill the drafter. To generate the draft for the drafter, at each timestep, the Attention Draft module accepts as the input the 4  representations of the current position (assuming position i) and its preceding positions (i-3, i-2, i-1)  generated by the last Transformer layer of the base model, and upsamples the input representations by 2 times  via copying the input representations. Then through the Transformer layer of the Attention Draft module, the output representations are fed to LM head. LM head will generate 8 raw draft tokens in a non-autoregressive way and the final draft tokens will be produced by CTC Transform module. The CTC-loss is used to count the probabilities of all the draft sequences that can be transformed into the ground truth sequence via dynamic programming and the sum of all these sequences is maximized as the training loss. In this way, the probability distribution is drawn towards the draft sequences that can derive ground truth sequence being allocated bigger probability. This means the candidate draft sequence with more reasonable sequence correlations will be selected as the winner at a greater likelihood.
> > >
> > > At inference, the base model does not generate the whole sequence in advance. Instead, at each timestep, it is used to generate the representations fed to the Attention Draft module and verify the generated draft tokens according to the probability by performing teacher forcing decoding to generate the draft tokens. If the probability to generate the draft tokens is greater than the set threshold, the base model will accept the draft tokens and uses them as the following generated tokens. Then the base model continues to encode these generated draft tokens in an autoregressive way. If the probability is smaller than the threshold, the base model will reject the draft tokens and generate the next token on its own and meanwhile encode the generated token in an autoregressive way, too.  Then the latest generated 4 representations by the last Transformer layer of the base model are used as the input of the Attention Draft module for the next timestep and the decoding goes to the next timestep. Here the Attention Draft module works in the same way as in training to generate input representations for LM head. Here LM head still generates N tokens for the next N positions in a non-autoregressive way with each position reserving top k tokens (k is a hyperparameter, which is set to 10 for current CTC-drafter), then the token sequence of the N positions with the highest probability will be selected as the raw draft via the token tree structure used in Medusa. The final draft generated via CTC transform will be fed to the base model to verify and the base model will decide to use the draft tokens as the next several output tokens or to generate the next token by itself where the former decision can generate several tokens at once and hence improves the decoding speed.

---

> ### Author Response · Authors · 2024-08-12
>
> Dear Reviewers,
>
> We are writing to kindly request your feedback.
>
> As the author-reviewer discussion phase is nearing its end, we are eagerly awaiting your response to address any questions or concerns you may have.
>
> We apologize for any inconvenience and appreciate your time and efforts.

---

> > ### Comment · Reviewer_NRvU · 2024-08-12
> > **Thank you**
> >
> > Thank you for further clarification.
> >
> > These detailed explanations make more sense than the information included in the original manuscript. In fact, this is quite "new" information to me (ex: 4 previous tokens, 2x copying, etc.) and possibly to other reviewers.
> >
> > I increased the soundness score from 2 to 3, and the overall score from 5 to 6.
> >
> > By the way, I wonder if that information was given in the original version. We might have asked different questions about architectural details. I note this to let the meta reviewer decide.

---

> > > ### Author Response · Authors · 2024-08-12
> > >
> > > Many thanks for your time and valuable suggestions to help improve our paper. We will follow your suggestions to include necessary clarifications in the revised version.

---

### Official Review · Reviewer_6n7r · 2024-07-14

**Soundness:** 3
**Presentation:** 2
**Contribution:** 3
**Rating:** 6
**Confidence:** 4

**Summary:**

The authors study the setup of speculative decoding where multiple tokens are being generated at parallel.
In this setup the authors use the idea of connectionist temporal classification to train a draft model which generates multiple tokens in parallel.

**Strengths:**

- The method shows significant improvement in token acceptance rates.

- The speedups reported are quite impressive.

**Weaknesses:**

- The speedups are unclear, it is calculated using the token acceptance rate rather than on a real system

- Details are not clear about at what layers the authors are taking the intermediate representation

- The details of training overhead are unclear

- It will be great if authors can compare to - Draft & Verify: Lossless Large Language Model Acceleration via Self-Speculative Decoding

**Questions:**

See weakness section

**Limitations:**

The authors have listed some weaknesses however it is not clear the cases where the idea will fail.

Specifically how dependent is data drift between training the draft model and inference.

---

> ### Author Rebuttal · Authors · 2024-08-06
>
> Many thanks for the insightful comments and constructive suggestions.
>
> $\textbf{R1. Evaluate the speedup on a real system(W1). }$
>
> We measured the speedup not only using the token acceptance rate(denoted as $\gamma$), but also the inference speedup which is conducted on a real system(denoted as $\beta$). In Section 4.1, we clarify the calculation of these two items in equation(12) and equation(13). For inference speedup $\beta$, we first record the average decoding time for each token of base model without speculation method ${\bar{T}}_{vanilla}$. Then measure the average time of models with different speculation methods
>
> ${\bar{T}}_{spec}$. The speedup is gained by dividing these two average values. All evaluations are conducted on the same device, which reflects performance of a real system. The speedup performance of different draft methods on MT-bench and GSM8K is displayed in Table 1. CTC-drafter shows superior speedup performance on a real system compared with Medusa and Hydra.
>
> $\textbf{R2. At what layers the intermediate representation are taken? (W2) }$
>
> All the intermediate representation are taking from the last transformer layer of base model which covers the complete hidden states. The intermediate representation is then transferred as the input of draft module to generate candidate sequences.
>
> $\textbf{R3. Details about the training overhead(for W3 and Q3). }$
>
> Similar to Figure 3 which displays the inference overhead, we give each stage’s time consumed throughout the whole training process as follow. Besides the conventional forward and back propagation which accounts for the main overhead, the calculation of CTC loss and label process are two additional steps that consume extra training time.
> | Forward Propagation | Label Process| CTC loss calculation| Back Propagation | Others |
> |:------------------|:--------------------:|:--------------------:|:--------------------:|:--------------------:|
> | 26.50%   |    6.01%      |    12.72%      |    54.06%      |    0.71%      |
>
> We follow the dynamic programming calculation of CTC loss to aggregate over all potential alignments that ultimately form the label sequence after CTC blank-collapse. To enable parallel label supervision, complete label sequences are sliced to short pieces and then be used in the calculation of CTC loss. These two extra stages are necessary when using CTC loss as training objective, causing additional training overhead.
>
> Besides, we list some main experimental environment settings in Section 4.1. In detail, for training task in Vicuna-7b, forward propagation and gradient upgrade were conducted on each device, following data parallelism. For Vicuna-13b, every two devices support one training process. For Vicuna-33b, all four devices are utilized together to conduct training following model parallelism. For base models of these three sizes, we uniformly set the max number of training epoch to 20. In practice, the models approach convergence after approximately 8 epochs. It takes around two days to train draft module for Vicuna-7b and Vicuna-13b. The training time consumed increase for Vicuna-33b, which is around four days.
>
>
> $\textbf{R4. Comparison with Draft and Verify (W4). }$
>
> Draft & Verify: Lossless Large Language Model Acceleration via Self-Speculative Decoding presents an impressive plug-and-play and cost-effective solution for inference acceleration with Skipped Layer and Draft-Exiting mechanism. It focuses on base model itself to draft tokens without additional draft module, which is distinct from our work.
> We have conducted evaluation experiments with llama-2-chat-13B as the base model and compared the speedup performance with Draft & Verify. The results are displayed as follow, with the temperature is set to 0. Due to time limitation, the results of evaluations on base models of other types like llama-2-13B will be included in our next version.
>
> | Draft method | Speedup $\beta$ |
> |:------------------|:--------------------:|
> | CTC-drafter   |    2.334×      |
> | Draft /& Verify       |    1.409×      |
>
> Since CTC-drafter introduces extra draft module compared with Draft & Verify, higher speedup performance can be viewed. However, it is a novel idea that considering skipped-layers for CTC-drafter. We will further explore possible improvement when introducing these mechanisms in our future work.

---

> ### Author Response · Authors · 2024-08-11
> **Supplementary Explanations by Authors**
>
> Thanks for your valuable suggestions. We reorganized the description of our method, hoping it can help understand our method better.
>
> Overall, the architecture of our method can be decomposed into three parts: the base model, the drafter which includes the Attention Draft module and LM head, and the CTC-related module which is used to calculate CTC-loss for training and performs CTC-style decoding at inference. The main structures of the three parts are as follows.
> - The base model in our method is based on the autoregressive generation framework like Llama which only uses Transformer decoder as the main structure.
> - For the drafter, the Attention Draft module employs one Transformer layer as its structure including masked multi-head self-attention sublayer and Feed Forward sublayer. Compared to Medusa which employs FFN as the structure of each Medusa head without interacting between Medusa heads, our method can introduce sequence correlations between the preceding generated words and next several words to generate, besides the CTC-related module.
> - The CTC-related module involves non-autoregressive generation of the next N tokens (N  is a hyperparameter, which is set to 4 for current CTC-drafter) where blank character and repetitive tokens are introduced into the raw generated sequence which will be processed by CTC Transform module to produce the final draft for the current decoding timestep.
>
> Our method works in a different way during training and inference. During training, the base model will generate the whole sequence in an autoregressive manner which is used as the ground truth to distill the drafter. To generate the draft for the drafter, at each timestep, the Attention Draft module accepts as the input the 4  representations of the current position (assuming position i) and its preceding positions (i-3, i-2, i-1)  generated by the last Transformer layer of the base model, and upsamples the input representations by 2 times  via copying the input representations. Then through the Transformer layer of the Attention Draft module, the output representations are fed to LM head. LM head will generate 8 raw draft tokens in a non-autoregressive way and the final draft tokens will be produced by CTC Transform module. The CTC-loss is used to count the probabilities of all the draft sequences that can be transformed into the ground truth sequence via dynamic programming and the sum of all these sequences is maximized as the training loss. In this way, the probability distribution is drawn towards the draft sequences that can derive ground truth sequence being allocated bigger probability. This means the candidate draft sequence with more reasonable sequence correlations will be selected as the winner at a greater likelihood.
>
> At inference, the base model does not generate the whole sequence in advance. Instead, at each timestep, it is used to generate the representations fed to the Attention Draft module and verify the generated draft tokens according to the probability by performing teacher forcing decoding to generate the draft tokens. If the probability to generate the draft tokens is greater than the set threshold, the base model will accept the draft tokens and uses them as the following generated tokens. Then the base model continues to encode these generated draft tokens in an autoregressive way. If the probability is smaller than the threshold, the base model will reject the draft tokens and generate the next token on its own and meanwhile encode the generated token in an autoregressive way, too.  Then the latest generated 4 representations by the last Transformer layer of the base model are used as the input of the Attention Draft module for the next timestep and the decoding goes to the next timestep. Here the Attention Draft module works in the same way as in training to generate input representations for LM head. Here LM head still generates N tokens for the next N positions in a non-autoregressive way with each position reserving top k tokens (k is a hyperparameter, which is set to 10 for current CTC-drafter), then the token sequence of the N positions with the highest probability will be selected as the raw draft via the token tree structure used in Medusa. The final draft generated via CTC transform will be fed to the base model to verify and the base model will decide to use the draft tokens as the next several output tokens or to generate the next token by itself where the former decision can generate several tokens at once and hence improves the decoding speed.

---

> ### Author Response · Authors · 2024-08-12
>
> Dear Reviewers,
>
> We are writing to kindly request your feedback.
>
> As the author-reviewer discussion phase is nearing its end, we are eagerly awaiting your response to address any questions or concerns you may have.
>
> We apologize for any inconvenience and appreciate your time and efforts.

---

> > ### Comment · Reviewer_6n7r · 2024-08-12
> > **Thank your for response**
> >
> > Appreciate the author response.
> >
> > Thank you for providing clarifications.
> >
> > I will bump up the score to 6.

---

> > > ### Author Response · Authors · 2024-08-12
> > >
> > > Many thanks for your time and valuable suggestions to help improve our paper. We will follow your suggestions to include necessary clarifications and experiments in the revised version.

---

### Official Review · Reviewer_EXv7 · 2024-07-22

**Soundness:** 2
**Presentation:** 3
**Contribution:** 2
**Rating:** 5
**Confidence:** 3

**Summary:**

The paper proposes a novel framework, CTC-drafter, to accelerate speculative decoding in large language models (LLMs). The authors introduce the use of Connectionist Temporal Classification (CTC) as a training objective, replacing the traditional cross-entropy loss. This method aims to improve context modeling and generate adaptive candidate sequences, which purportedly enhances the speed and accuracy of the speculative decoding process. The paper demonstrates the effectiveness of CTC-drafter through experiments on various benchmarks and compares its performance with existing methods such as Medusa and Hydra.

**Strengths:**

- Originality: Introducing CTC as a training objective for speculative decoding is a novel approach.

- Quality: The theoretical framework is well-developed, with clear definitions and derivations.

- Clarity: The paper is generally well-written and logically structured, making it accessible to a broad audience.

**Weaknesses:**

- Experimental Validation: The experiments do not fully validate the claims. There is a need for more extensive testing across different datasets and model architectures to ensure the generality and robustness of the proposed method.

- Comparative Analysis: While comparisons are made with Medusa and Hydra, the analysis lacks depth. More detailed insights into why

- CTC-drafter performs better or worse in specific scenarios would be beneficial.

**Questions:**

1. Can the authors provide more details on the hyperparameter settings and training configurations used in the experiments?

2. How does the performance of CTC-drafter vary with different model architectures and dataset sizes?

3. Can the authors elaborate on the computational overhead introduced by the CTC loss and how it compares to the benefits in inference speed?

**Limitations:**

The authors have discussed several limitations, including the need for more training tricks to enhance the draft module and the uncertainty regarding the optimality of the current draft model structure.

---

> ### Author Rebuttal · Authors · 2024-08-06
>
> Many thanks for the insightful comments and constructive suggestions.
>
> $\textbf{R1. More details on the hyperparameter settings and training configurations used in the experiments (Q1). }$
>
> The main training configurations are listed in Section 4.1 including learning rate, gradient clipping threshold and the max length of training data, which is relatively crucial to the reproduction of experiment results. In this part we clarify other hyperparameters involved in the training of CTC-drafter. The total training number of epoch is set to 20. Uniform noise is added to enhance the robustness of CTC-drafter. AdamW optimizer is selected with $\beta_{1} = 0.9$ and  $\beta_{2} = 0.95$. For the settings of CTC loss, we choose the token <unk> in the vocabulary to represent blank token $\epsilon$ of CTC algorithm.We will add these details in the next version.
>
> $\textbf{R2. How does the performance of CTC-drafter vary with different model architectures and dataset sizes?(Q2) }$
>
> Regarding your concern about the different model structures, we would appreciate more clarification. Are you referring to the base model that requires acceleration, or the draft model used for speculation? To cover all bases, we provide an explanation for both aspects.
>
> For base model, we evaluate the performance in Vicuna-7b, Vicuna-13b and Vicuna-33b in Table 1. The architecture of base model is the same as Medusa and Hydra for fair comparison. CTC-drafter outperforms other draft models in base models of all sizes. We also added evaluation experiments on MT-bench with LLaMA2-Chat as base models to ensure the generality and robustness. The results are displayed as follow. CTC-drafter keeps a desirable speedup performance when transferring to LLaMA2-Chat-7b. The results on LLaMA2-Chat-13b and LLaMA2-Chat-70b will be included in our next version.
>
> | Base models | Speedup $\beta$ | Token acceptance rate $\gamma$ |
> |:------------------|:--------------------:|:-------------------:|
> | Vicuna-7b      | 2.99×                          | 3.73                                  |
> | Vicuna-13b     | 2.52×         | 3.70                         |
> | Vicuna-33b     | 2.20×         | 3.53                         |
> | LLaMA2-Chat-7b    | 2.33×         | 3.24                         |
>
> For draft model, we modify the components to construct different structures and conduct evaluation in ablation experiment. The results are demonstrated in Table 2. It shows that with CTC loss and CTC blank-collapse, draft module is guided to conduct attention across the whole input sentence instead of simply learning offsets of the last hidden states. For fair comparison, the training dataset we choose is still ShareGPT, which contains 68000 dialogues and is also used in Medusa and Hydra. However, we introduce a new evaluation dataset GSM8K beside MT-bench to further elaborate the validity of CTC-drafter. The results on different evaluation datasets are recorded in Table 1. CTC-drafter achieves desirable speedup performance on both datasets, showing its generality.
>
> $\textbf{R3. Comprehensive and in-depth comparative analysis with Medusa and Hydra (W2). }$
>
> Based on the observation that currently the widely used draft models usually generate draft tokens for the next several positions in a non-autoregressive way without considering the correlations between draft tokens, we introduce CTC modeling strategy into draft model to strengthen the correlations between draft tokens during the draft phase, thereby generating higher-quality draft candidate sequences. Main experiment results on MT-bench and GSM8K (Table 1) reflect the performance improvement compared with Medusa and Hydra. Higher draft accuracy contributes to better speedup while the speedup performance of all speculation method is influenced by the larger ability gap as base model size increases.
>
> To further evaluate performance in specific scenarios, we further explore how speedup varies on different question categories, showing in Figure 2. CTC-drafter is relatively difficult to deal with Roleplay question, which may be due to the deficiency of questions of this category in our training datasets.
>
> Compared with Medusa and Hydra, it is unavoidable that our methods’ draft strategy requires more complex calculations. We display each stage’s time consumed throughout the whole inference decoding process in Figure 3. Although extra time added in one single decoding step, the overall decoding steps are decreased for better draft quality. It is acceptable to increase the draft ability and thus reducing base model’s decoding rounds, which balances the extra time consumption and achieve better speedup on the whole.
>
> Above is the comprehensive analysis based on the experiments for our work. If you have any suggestions for additional aspects that could be included, we would be most grateful to hear them and would be delighted to incorporate them in the next version.
>
> $\textbf{R4. The computational overhead introduced by CTC loss and how it compares to the benefits in inference speed? (Q3). }$
>
> We would like to clarify that the time-consuming CTC loss calculation is conducted only during training, while not at inference. When inference, extra computational overhead related to CTC includes first removing consecutive duplicate tokens and blank character ϵ and then modifying attention map used in base model verification. The detailed computational overhead at inference is elaborated in Figure 3. The computational overhead introduced by CTC discussed above is noted as CTC transform in the Figuren3. Compared with Medusa,
> the drafting of our method takes more time while it provides a better draft and leads to a higher acceptance rate by base model. Considering that the calculation of base model still accounts for the main overhead, our method achieves better speedup on the whole.

---

> > ### Comment · Reviewer_EXv7 · 2024-08-09
> >
> > Given that the experiments are indeed extensive, I will increase my score from 4 to 5.

---

> ### Author Response · Authors · 2024-08-11
> **Supplementary Explanations by Authors**
>
> Thanks for your valuable suggestions. We reorganized the description of our method, hoping it can help understand our method better.
>
> Overall, the architecture of our method can be decomposed into three parts: the base model, the drafter which includes the Attention Draft module and LM head, and the CTC-related module which is used to calculate CTC-loss for training and performs CTC-style decoding at inference. The main structures of the three parts are as follows.
> - The base model in our method is based on the autoregressive generation framework like Llama which only uses Transformer decoder as the main structure.
> - For the drafter, the Attention Draft module employs one Transformer layer as its structure including masked multi-head self-attention sublayer and Feed Forward sublayer. Compared to Medusa which employs FFN as the structure of each Medusa head without interacting between Medusa heads, our method can introduce sequence correlations between the preceding generated words and next several words to generate, besides the CTC-related module.
> - The CTC-related module involves non-autoregressive generation of the next N tokens (N  is a hyperparameter, which is set to 4 for current CTC-drafter) where blank character and repetitive tokens are introduced into the raw generated sequence which will be processed by CTC Transform module to produce the final draft for the current decoding timestep.
>
> Our method works in a different way during training and inference. During training, the base model will generate the whole sequence in an autoregressive manner which is used as the ground truth to distill the drafter. To generate the draft for the drafter, at each timestep, the Attention Draft module accepts as the input the 4  representations of the current position (assuming position i) and its preceding positions (i-3, i-2, i-1)  generated by the last Transformer layer of the base model, and upsamples the input representations by 2 times  via copying the input representations. Then through the Transformer layer of the Attention Draft module, the output representations are fed to LM head. LM head will generate 8 raw draft tokens in a non-autoregressive way and the final draft tokens will be produced by CTC Transform module. The CTC-loss is used to count the probabilities of all the draft sequences that can be transformed into the ground truth sequence via dynamic programming and the sum of all these sequences is maximized as the training loss. In this way, the probability distribution is drawn towards the draft sequences that can derive ground truth sequence being allocated bigger probability. This means the candidate draft sequence with more reasonable sequence correlations will be selected as the winner at a greater likelihood.
>
> At inference, the base model does not generate the whole sequence in advance. Instead, at each timestep, it is used to generate the representations fed to the Attention Draft module and verify the generated draft tokens according to the probability by performing teacher forcing decoding to generate the draft tokens. If the probability to generate the draft tokens is greater than the set threshold, the base model will accept the draft tokens and uses them as the following generated tokens. Then the base model continues to encode these generated draft tokens in an autoregressive way. If the probability is smaller than the threshold, the base model will reject the draft tokens and generate the next token on its own and meanwhile encode the generated token in an autoregressive way, too.  Then the latest generated 4 representations by the last Transformer layer of the base model are used as the input of the Attention Draft module for the next timestep and the decoding goes to the next timestep. Here the Attention Draft module works in the same way as in training to generate input representations for LM head. Here LM head still generates N tokens for the next N positions in a non-autoregressive way with each position reserving top k tokens (k is a hyperparameter, which is set to 10 for current CTC-drafter), then the token sequence of the N positions with the highest probability will be selected as the raw draft via the token tree structure used in Medusa. The final draft generated via CTC transform will be fed to the base model to verify and the base model will decide to use the draft tokens as the next several output tokens or to generate the next token by itself where the former decision can generate several tokens at once and hence improves the decoding speed.

---

> ### Author Response · Authors · 2024-08-12
>
> Dear Reviewers,
>
> We are writing to kindly request your feedback.
>
> As the author-reviewer discussion phase is nearing its end, we are eagerly awaiting your response to address any questions or concerns you may have.
>
> We apologize for any inconvenience and appreciate your time and efforts.

---

### Decision · Program_Chairs · 2024-09-25

**Decision:**

Accept (poster)

**Comment:**

This paper applies the Connectionist Temporal Classification (CTC) approach to draft model training and inference in speculative decoding. By incorporating CTC, the draft model considers all possible candidate sequences that could generate the target during training, leading to better candidate generation during inference and a higher acceptance rate. The proposed method demonstrates significant speedup compared to previous approaches across several benchmarks. All reviewers agreed that this paper is above the acceptance threshold.